# Techno-Economic Evaluation of Hand Sanitiser Production Using Oil Palm Empty Fruit Bunch-Based Bioethanol by Simultaneous Saccharification and Fermentation (SSF) Process

**Andre Fahriz Perdana Harahap [1]** , **Jabosar Ronggur Hamonangan Panjaitan [2]**,
**Catia Angli Curie [1]**, **Muhammad Yusuf Arya Ramadhan [1]**, **Penjit Srinophakun [3]** and
**Misri Gozan [1],\***

[1] Chemical Engineering Department, Faculty of Engineering, Universitas Indonesia, Depok 16424, Indonesia; andrefahriz25@gmail.com (A.F.P.H.); catiacurie@gmail.com (C.A.C.); yra.ramadhan@gmail.com (M.Y.A.R.)

[2] Chemical Engineering Program, Institut Teknologi Sumatera, Lampung 35365, Indonesia; jabosarronggur@gmail.com

[3] Chemical Engineering Department, Faculty of Engineering, Kasetsart University, Bangkok 10900, Thailand; fengpjs@ku.ac.th

\* Correspondence: mgozan@che.ui.ac.id; Tel.: +62-857-811-76292

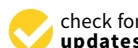

**Featured Application: The results of this work can be applied as the basis of consideration during the feasibility study step of plant design supposed to produce biomass-based bioethanol for commercial-scale hand sanitiser production. The results can contribute in the effort of converting agricultural waste into more valuable alternative product such as hand sanitiser which is urgently needed in global pandemic situation.**

**Abstract:** Oil palm empty fruit bunch (OPEFB) is a potential raw material abundantly available for bioethanol production. However, the second-generation bioethanol is still not yet economically feasible. The COVID-19 pandemic increases the demand for ethanol as the primary ingredient of hand sanitisers. This study evaluates the techno-economic feasibility of hand sanitiser production using OPEFB-based bioethanol. OPEFB was alkaline-pretreated, and simultaneous saccharification and fermentation (SSF) was then performed by adding *Saccharomyces cerevisiae* and cellulose enzyme. The cellulose content of the OPEFB increased from 39.30% to 63.97% after pretreatment. The kinetic parameters of the OPEFB SSF at 35 °C, which included a $\mu$ max, ks, and kd of 0.018 $h^{-1}$, 0.025 $g/dm^3$, and 0.213 $h^{-1}$, respectively, were used as input in SuperPro Designer® v9.0. The total capital investment (TCI) and annual operating costs (AOC) of the plant were \$645,000 and \$305,000, respectively, at the capacity of 2000 kg OPEFB per batch. The batch time of the modelled plant was 219 h, with a total annual production of 32,506.16 kg hand sanitiser. The minimum hand sanitiser selling price was found to be \$10/L, achieving a positive net present value (NPV) of \$108,000, showing that the plant is economically feasible.

**Keywords:** bioethanol; economic analysis; hand sanitiser; oil palm empty fruit bunch (OPEFB); simultaneous saccharification and fermentation; SuperPro Designer®

## 1. Introduction

The World Health Organization (WHO) declared the outbreak of novel coronavirus SARS-CoV-2 as a global pandemic because of its ease of spread, severity it may cause and the lack of global

precaution back then [1]. SARS-CoV-2 can be easily transmitted among humans through droplets, and human-coronaviruses can remain active on surfaces for up to nine days [2]. People may also be infected without showing severe symptoms. Practising cough and sneeze etiquette, as well as hand hygiene, is thus encouraged as a preventive measure. Following the WHO pandemic declaration, many countries then put in place strong restrictions and even lockdown to urge their citizens to limit physical contact and slow down the spread of Covid-19 as much as possible. However, as of June 2020, countries are beginning to relax the lockdown and restrictions because of economic demand. Since a vaccine for COVID-19 is yet to be developed, individual precautions must be practised in a more disciplined manner to prevent the spread of the virus. These have made hand sanitiser one of the most crucial items to have on hand.

Hand sanitiser helps the achievement of hand hygiene, especially in clinical settings or in conditions where water is unavailable. A study performed by Rai et al. has shown that single-use hand sanitiser packets are quicker to use than single-use moist towelettes. Thus, hand sanitisers are often preferred [3]. However, most hand sanitisers are not effective for non-enveloped viruses. Thus, despite being practical, hand sanitiser cannot fully replace hand washing, especially when the hand is visibly dirty or has come into contact with harmful chemicals [4–6].

Hand sanitisers can be classified into two groups: alcohol-based and alcohol-free sanitisers. Alcohol-based hand sanitisers are more common because of the ease of preparation, low cost and efficacy. The WHO has released a guide for the local production of alcohol-based hand-rub formulations. Generally, the alcohol used can be ethanol, isopropyl alcohol, or n-propanol. An alcohol content of 60–95% *v/v* is required for alcohol-based sanitisers to kill microbes effectively. Alcohols render microbes ineffective by damaging their lipid membranes and/or denaturing the proteins. The WHO-recommended formulations contain either 80% *v/v* of ethanol or 75% *v/v* of isopropyl alcohol with 1.450% *v/v* glycerol, 0.125% *v/v* hydrogen peroxide and water. Compared to isopropanol, ethanol is less irritant to the skin and more effective in killing a broader range of microbes [5–7].

Ethanol is commonly produced either from petroleum feedstock or from a sugar/starch feedstock, which is referred to as first-generation bioethanol. Neither is sustainable because petroleum is non-renewable while using sugar or starch as a feedstock interferes with food supplies and the utilisation of fertile land. Hence, the use of lignocellulosic feedstock, second-generation bioethanol, offers greater potential in terms of sustainability. This is because lignocellulosic materials are derived from various organic waste products and residues, including agricultural, forestry and household waste. Lignocellulose generally contains cellulose (40–60%), hemicellulose (20–40%) and lignin (10–25%) with the exact content depends on the biomass source and harvest time. To produce ethanol, lignocellulosic feedstocks must undergo four basic steps: pretreatment, hydrolysis, fermentation and purification. The four steps may be assembled using the separate hydrolysis and fermentation (SHF) process, the simultaneous saccharification and fermentation (SSF) process, the simultaneous saccharification and co-fermentation (SSCF) process or consolidated bioprocessing (CBP). In SHF, each step stands alone. Thus, each can be performed under its optimum conditions. However, the capital cost of the SHF is high. In SSF, the sugar released from the saccharification process is directly fermented in the same reactor, limiting the risk of inhibition. SSF is currently the process used most often. Yet, saccharification and fermentation have different optimum conditions. Thus, it is still a challenge to perform both processes optimally in one reactor. SSCF differs from SSF in that it allows for the simultaneous fermentation of all the sugars produced from cellulose and hemicellulose. CBP, in turn, is different from the other processes because it uses only one type of microorganism to produce the enzymes for hydrolysis and to obtain fermentation. This process offers the most efficient process as it combines enzyme production, enzymatic hydrolysis and fermentation in one pot. Yet, the development of the CBP process is dependent upon the ability to engineer the appropriate microbes [8–10].

Many studies have been conducted to assess the production of bioethanol from various lignocellulose sources, such as from sugarcane bagasse using the SSF method and pretreated with

white-rot fungi, from durian skin using the SSF method and cell encapsulation of *S. cerevisiae*, from paper waste using the SSF method, from corncobs using the SSCF method and scaled up to demo scale and from sugarcane bagasse using the CBP process [11–16]. Because of the complexity of the process, to date, lignocellulosic ethanol is still more expensive to produce compared to first-generation bioethanol. The productions of second-generation bioethanol are still covering less than 3% of total global bioethanol production where they are limited only at negligible amount in some demo plants around the world that work industrially but are not yet economically feasible [17,18]. One way of tackling this challenge is to produce various by-products in lignocellulosic ethanol plants. Rosales-Calderon and Arantes have reviewed several biochemicals with a minimum technology-readiness level of eight that can be produced alongside lignocellulosic ethanol [9]. The production of hand sanitisers can also be an option because of the ease of production and the increasing demand.

Oil palm empty fruit bunches (OPEFBs) are solid waste lignocellulosic biomass products of the oil palm industry, consisting of cellulose, hemicellulose and lignin. As much as 1.1 tonnes of OPEFBs can be produced from processing 1 tonne of palm oil. In 2011, the global production of OPEFBs amounted to 14.5 million tonnes, of which half were produced in Indonesia [19–21]. Their cellulose, hemicellulose and lignin contents give OPEFBs the potential to be converted into various biochemicals, such as bioethanol, furfural, formic acid and levulinic acid. Sahlan et al. have examined bioethanol production from OPEFBs using *Rhizopus oryzae* encapsulated using calcium alginate [14]. Panjaitan and Gozan have investigated formic acid production from the acid catalysed hydrolysis reaction of OPEFBs; Harahap et al. have further optimised the production process using the response surface methodology (RSM) method [22,23]. In addition, Gozan et al. have produced levulinic acid and furfural from OPEFBs and have evaluated its production kinetics [24].

As a demand and sustainability, palm oil serves many tasks such as excellent source of biomass, self-sufficient energy in processing, effective carbon sink, positive contribution to energy balance, sustainable practices and zero burning. The Indonesian Sustainable Palm Oil (ISPO) Certification Scheme is introduced by the Government of Indonesia as a mandatory requirement for all oil palm growers and mills to enhance the competitiveness of Palm Oil in the global market. For Malaysia, The Malaysian Sustainable Palm Oil (MSPO) Certification Scheme is the national scheme in Malaysia for oil palm plantations, independent and organised smallholdings, and palm oil processing facilities to be certified against the requirements of the MSPO Standards. The MSPO Certification Scheme provides the general principles for plantations and palm oil processing facilities to ensure that the palm oil products are produced in a responsible and sustainable manner [25].

Process simulation can be used to gain a better understanding of particular processes and their behaviours in real life. Simulation can be used to conduct experiments to evaluate behaviours or alternatives to chemical process systems [26]. To simulate a process, process engineering software needs specific scientific abilities, including, but not limited to, the ability to accurately describe the physical properties of pure components and complex mixtures, the ability to model a large variety of reactors and unit operations, and the computational and numerical techniques needed to solve a wide variety of different mathematical equations [27].

After thoroughly analysing a chemical process system, an economic analysis is an important step in developing better knowledge about that process. There are several parameters used to measure the investment performance of the production of chemical products, including, but not limited to, the internal rate of return (IRR), the net present value (NPV), and the payback period (PBP). The net present value is the difference between the present value of cash inflows and outflows used to analyse the profitability of the project. The internal rate of return is a value representing the profitability potential of the project as a form of discount rate that reduces the NPV of the project to zero. The payback period is the amount of time needed to recover the initial investment from the project. The rate of investment (ROI) is a percentage showing the efficiency of a project calculated from the net profit divided by the total cost of investment [28].

Advances in computational technology and improvements in modelling techniques have enabled easier access to quantitative predictions of complex systems' behaviours; in particular, mass and energy balances could, with further implementation, be used to accurately design unit operations. Process simulation has been commonly utilised in real-world chemical engineering industries to perform system analysis and synthesis [29]. Some of the leading commercial process simulation software packages today include Aspen Plus®, Aspen HYSYS®, CHEMCAD and PRO/II with PROVISION. The above simulators have been designed to model primarily continuous processes and their transient behaviour for process control purposes. Most bioprocess and biorefinery products, however, are produced in batch and semi-continuous modes. Such processes are best modelled with batch process simulators that account for time-dependencies and the sequencing of events. It is not difficult to switch between simulators once the principles of process simulation have been understood by the engineers [30]. SuperPro Designer® v.9.0. (Intelligen, Inc., Scotch Plains, NJ, USA, 2019) is one of the simulators specifically developed to simulate batch process systems. SuperPro Designer® also has the capability to perform comprehensive economic analyses, which is not limited to calculating total capital investment, NPV, IRR and PBP but can also perform detailed calculations of operational expenditure (OPEX) by enabling the customisation of staffing, utility pricing, and process scheduling. Therefore, SuperPro Designer® is perfect for the simulation needed in this study.

Until now, there have been no articles yet addressing the economic feasibility to produce value-added product like hand sanitiser, which is globally needed in the current pandemic situation, from under-utilised resource like OPEFBs. This study aims to provide an alternative application for second-generation bioethanol, combining both laboratory and modelling studies. A hand sanitiser plant utilising OPEFB as its feedstock was designed, and the kinetic parameters of the production of bioethanol from OPEFB using the SSF method were analysed. The obtained parameters were then used in the designed process simulation using SuperPro Designer® to evaluate the economic viability of this hand sanitiser plant.

## 2. Materials and Methods

### 2.1. Kinetic Evaluation of Fermentation

#### 2.1.1. Preparation of Materials

The OPEFB used as the raw material for bioethanol production in this study was kindly received from the Indonesian government-owned palm oil plantation company PTPN (PT Perkebunan Nusantara) VIII Kertajaya, located in Banten, Indonesia. Before being used, the OPEFB was first reduced in size by cutting and grinding it in a laboratory-scale wood grinder (PT. Enerba Teknologi, Tangerang Selatan, Banten, Indonesia). The obtained OPEFB fibre was sifted through a 40-mesh stainless steel wire filter in order to obtain a uniform size of raw material, as shown in Figure 1. The OPEFB fibre was then washed under clean running water to remove the attached dirt and finally dried in an oven at 80 °C for 24 h. Cellulase enzymes from *Trichoderma reesei* ATCC 26921 and yeast *Saccharomyces cerevisiae* Type-1 were purchased from Sigma Aldrich (St. Louis, MO, USA). Both the cellulase and the yeast were stored in a refrigerator at 10 °C before being used. All other analytical-grade chemical reagents, like citrate buffer and sodium hydroxide, were purchased from Merck (Darmstadt, Hesse, Germany).

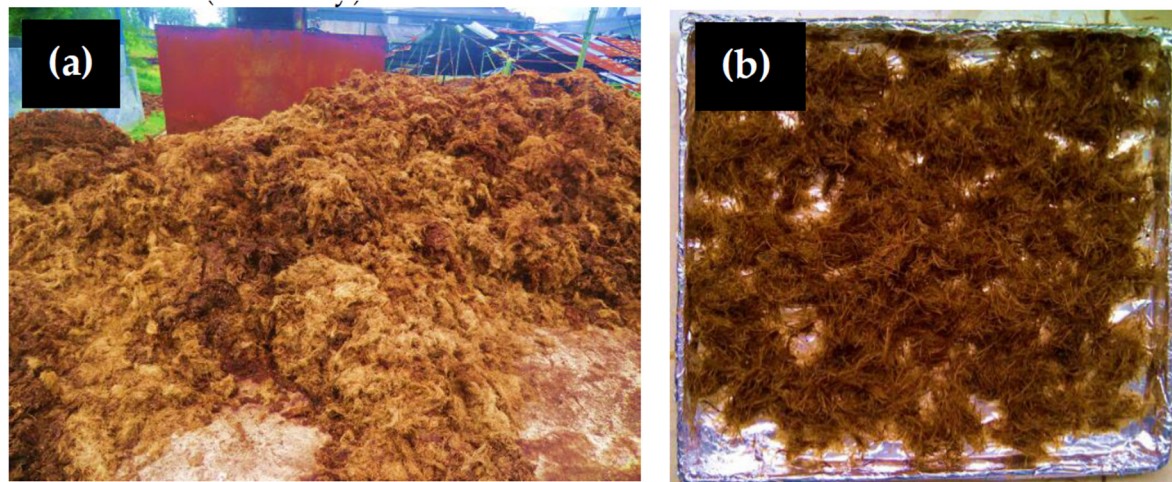

**Figure 1.** OPEFB before (**a**) and after (**b**) being ground as raw material for bioethanol production.

### 2.1.2. OPEFB Pretreatment and Analysis

OPEFB that will be used as a raw material for SSF must first be pretreated to remove its lignin content, thus enhancing the accessibility of the cellulase enzyme to the cellulose for further bioethanol conversion from glucose [31]. In this study, an alkaline pretreatment method using sodium hydroxide as the solvent was applied, according to Zulkiple et al. [32]. Pretreatment was performed by adding 30 g of the OPEFB sample into a 10% sodium hydroxide solution with a 1:10 solid-to-liquid ratio. The mixture was then heated to 120 °C for 2 h in an autoclave reactor (CV. Pugar Mandiri Teknik, Bandung, West Java, Indonesia) and cooled down at room temperature. The pretreated OPEFB was separated from the solvent, known as black liquor, using a stainless-steel wire filter. It was then washed with clean hot water to remove any remaining sodium hydroxide until neutrality was attained and dried in an oven at 95 °C for 24 h. The pretreated OPEFB was then analysed for its cellulose, hemicellulose, lignin and ash contents according to SNI 0444:2009, SNI 14-1304-1989, SNI 0492:2008 and SNI 0442:2009, respectively [33–36]. At this stage, the OPEFB was ready to be used for bioethanol production using the SSF method.

### 2.1.3. Simultaneous Saccharification and Fermentation

The SSF medium consisted of 15 g of pretreated OPEFB, 200 mL of 0.05 M citrate buffer (pH 4.8), 1 mL of cellulase enzyme and 5 g of dry yeast. The mixture of OPEFB and citrate buffer was first sterilised at 121 °C for 20 min in an autoclave. The mixture was then cooled down to room temperature and added to both the cellulase enzyme and the dry yeast. The SSF was performed in an Erlenmeyer flask (DWK Life Sciences GmbH, Mainz, Germany) closed tightly with aluminium foil. The flask was then incubated in an orbital shaker (Thermo Fisher Scientific, Waltham, MA, USA) at 150 rpm for 96 h. These experiments were conducted for temperature variations of 30 °C, 32 °C and 35 °C to give sufficient data for the determination of the kinetic parameters. Samples were taken at 24, 48, 72 and 96 h to analyse the concentrations of glucose, yeast cells and bioethanol in the fermentation broths.

### 2.1.4. Analysis of Fermentation Products

The fermented cell and glucose concentrations were analysed using optical density and dinitrosalicylic acid (DNS) methods. The optical density and DNS analyses were performed using UV-vis spectroscopy (UV-M90, BEL Engineering srl ©, Monza, MB, Italy). The DNS analysis used a 575 nm wavelength; the blank solution was a DNS buffer citrate solution. The optical density analysis used a 600 nm wavelength; the blank solution was a citrate buffer solution. The bioethanol analysis

was conducted using a SUPELCOWAX-10 gas chromatography analyser (Supelco Inc., Bellefonte, PA, USA) at 50 °C.

### 2.1.5. Reaction Constants Determination

The kinetic model of bioethanol production used in this study was chosen in accordance with Fogler (2004), under the assumption that there is no mass-transfer effect and that the influence of bioethanol as an inhibitor can be ignored [37]. The kinetic equation can, therefore, be simplified to:

- The kinetic equation for yeast cell formation:

$$\frac{dC_C}{dt} = \left( \mu \max \frac{Cc \cdot Cs}{ks + Cs} - kd \cdot Cc \right) \tag{1}$$

- The kinetic equation for product formation (bioethanol):

$$\frac{dC_C}{dt} = Y_{\frac{P}{C}} \cdot \mu \max \frac{Cc \cdot Cs}{ks + Cs} \tag{2}$$

- The kinetic equation for the residual substrate:

$$\frac{dC_s}{dt} = Y_{\frac{S}{C}} \cdot \left( \mu \max \frac{Cc \cdot Cs}{ks + Cs} \right) - m \cdot Cc \tag{3}$$

From the equations, it is necessary to determine the four reaction rate constants ($\mu \max$, $ks$, $kd$ and $m$). Experimental data were obtained for the cell, glucose and bioethanol concentrations over time, and the reaction rate constants could then be predicted by minimising the sum of the square error values between the experimental data and the prediction data that was formulated:

$$SS = \sum_i \sum_j \left( [j]_{i\ exp} - [j]_{i\ predicted} \right)^2 \tag{4}$$

The sum of the square error value can be determined using fminsearch optimisation (the Nedler-Mead method) and MATLAB software (MATLAB 9.6, MathWorks, Natick, MA, USA, 2019).

### 2.2. Hand Sanitiser Plant Design

SuperPro Designer® v.9.0 was used to conducting the process simulation and economic assessment of this hand sanitiser plant. Process design should always be completed first in order to specify a step-by-step process as well as to determine the equipment that will be involved in the simulation. At this stage, users also input specific parameters and operating conditions directly in the software for each involved piece of equipment. To model this hand sanitiser plant, a capacity of 2000 kg per batch of OPEFB was determined. The plant is also assumed to be constructed inside a palm oil mill area to reduce the transportation cost associated with the OPEFB.

The first section of this hand sanitiser plant is the pretreatment section. In this section, OPEFB obtained from the palm oil mill is first reduced in size by a grinding machine. This equipment grinds the raw material from the palm oil mill into 40-mesh sized OPEFB for 1 h. The size-reduced OPEFB is then transferred into a vertical-on-legs tank, where the alkaline pretreatment takes place at 120 °C for 60 min. The OPEFB is mixed with a 10% sodium hydroxide solution with a 1:10 solid-to-liquid ratio by means of an adjustable mixer before entering this tank. The mixture is then cooled in a cooler for 60 min from 120 °C to 25 °C before being washed. During the washing process, the OPEFB is washed with 0.02 m³/kg of water for 30 min. This washing process is assumed to remove 60% of the water and 100% of the sodium hydroxide as aqueous waste known as black liquor. Some hemicellulose, lignin, and ash are also removed in this process, according to the pretreatment compositional analysis results.

The second section of the plant is devoted to medium preparation and SSF. The pretreated OPEFB from the previous washing process is mixed with a citrate buffer (0.380% citric acid, 0.720% sodium citrate and water) at 1:13.13 liquid-to-solid ratio by means of an adjustable mixer. The mixture is then sterilised using a heat steriliser at 120 °C for 2 h to avoid contamination from other organisms and cooled back down to 35 °C. Cellulase and yeast at 0.5% and 1% of the mass ratio of the entering feed, respectively, are gradually added to the mixture by means of two serial adjustable mixers. The mixture is then finally ready to be transferred into a batch vessel fermenter operating at 35 °C and equipped with a jacket for chilled water flow. SSF takes place in this fermenter, with three reactions occurring simultaneously. Equation (5) is related to the saccharification reaction. Equations (6) and (7) are related to ethanol production and cell formation, respectively. The batch time for SSF in this fermenter was set in such a way as to obtain the highest amount of bioethanol.

$$1.00 \text{ cellulose } + 13.13 \text{ water } \rightarrow 2.21 \text{glucose} \tag{5}$$

$$1.00 \text{ glucose } \rightarrow 2.09 \text{ carbon dioxide } + 1.91 \text{ethyl alcohol} \tag{6}$$

$$0.56 \text{ glucose } \rightarrow 0.45 \text{ carbon dioxide } + 1.11 \text{ water } + 0.33 \text{ yeast} \tag{7}$$

Reaction rate constants obtained from the optimisation results will be used as values to be input in the SuperPro Designer® v9.0 Software. The kinetics parameters data input can be seen in Table 1 below.

**Table 1.** Kinetics parameters data used as inputs in SuperPro Designer®.

| Reaction | Kinetics Parameters Input |
|---|---|
| Glucose → Yeast cell | $\frac{dC_C}{dt} = \left[ \mu \max \left[ 1 - \frac{Cp}{Cp^*} \right] \frac{Cs}{ks+Cs} - kd \right] Cc$<br>Input:<br>$\alpha = 1$<br>$\mu \max = \mu \max$<br>$(S1 - \text{Term}) = 1$<br>$(S2 - \text{Term}) = \frac{Cs}{ks+Cs}$<br>$\beta = -kd$<br>$(B - \text{Term}) = Cc$ |
| Glucose → Ethanol | $\frac{dC_C}{dt} = \left[ Y_{\frac{P}{C}} \cdot \mu \max \left[ 1 - \frac{Cp}{Cp^*} \right] \frac{Cs}{ks+Cs} \right] Cc$<br>Input:<br>$\alpha = Y_{\frac{P}{C}}$ (ethanol mass/cell mass)<br>$\mu \max = \mu \max$<br>$(S1 - \text{Term}) = 1$<br>$(S2 - \text{Term}) = \frac{Cs}{ks+Cs}$<br>$\beta = 0$<br>$(B - \text{Term}) = Cc$ |
| Glucose → Glucose(residue) | $\frac{dC_s}{dt} = \left[ Y_{\frac{S}{C}} \cdot \left( \mu \max \left[ 1 - \frac{Cp}{Cp^*} \right] \frac{Cs}{ks+Cs} \right) - m \right] Cc$<br>Input:<br>$\alpha = Y_{\frac{S}{C}}$<br>$\mu \max = \mu \max$<br>$(S1 - \text{Term}) = 1$<br>$(S2 - \text{Term}) = \frac{Cs}{ks+Cs}$<br>$\beta = -m$<br>$(B - \text{Term}) = Cc$ |

The final section of the plant handles purification and hand sanitiser formulation. The purification is started with filtration for 2 h in a plate and frame filter, which is set to remove almost all solid fractions from the fermentation broth. The liquid fraction is then transferred into the first distillation column, called a beer column, for 6 h to separate the ethanol from the fermentation medium. Most of the water and fermentation medium are removed as bottom products. At the same time, concentrated

ethanol is transferred to the second distillation column, called a rectifying column, to increase the ethanol concentration up to 96%. The product is then cooled to 25 °C and transferred into a blending tank, where 3% hydrogen peroxide, 98% glycerol and water are added to fulfil the hand sanitiser requirements provided by the WHO. The mixture is agitated for 1 h to obtain a homogenous hand sanitiser formulation and is then transferred out as a final product. The complete process flow diagram for the described hand-sanitiser production process as formulated by SuperPro Designer® v9.0 can be seen in Figure 2 below.

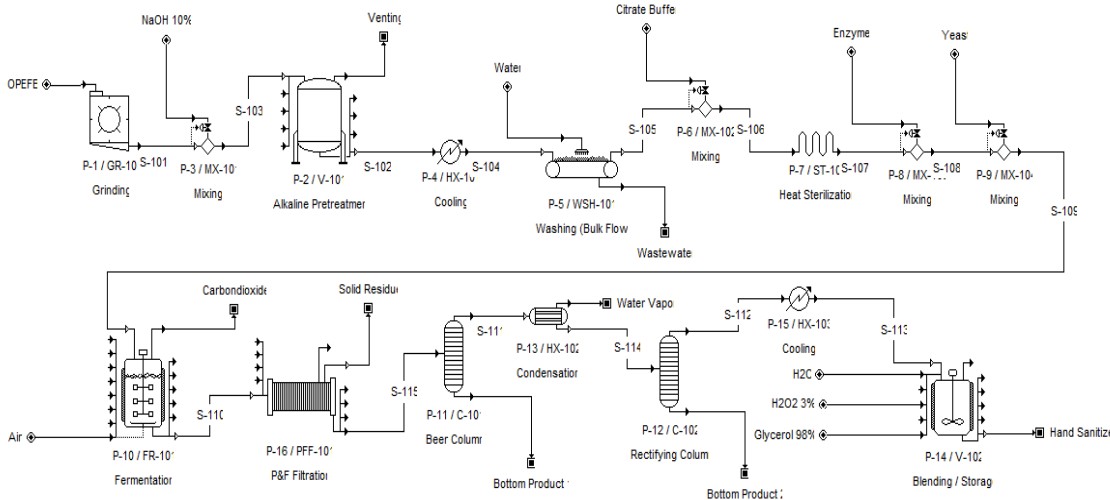

**Figure 2.** Complete process flow diagram of hand sanitiser production from OPEFB-based bioethanol in SuperPro Designer® v9.0.

The economic evaluation of this hand sanitiser plant was focused on the internal rate of return (IRR), the net present value (NPV), the payback period (PBP), the return on investment (ROI) and the gross margin of the operation. The calculations of those parameters were performed automatically by SuperPro Designer® v9.0. Before performing the economic evaluation, several terms should be assigned to this hand sanitiser plant, as can be seen in Table 2 below.

**Table 2.** Assigned terms for hand sanitiser plant's economic evaluation.

| Parameters | Value |
| --- | --- |
| Year of analysis | 2020 |
| Construction period | 30 months |
| Start-up period | 4 months |
| Project lifetime | 15 years |
| Interest rate | 7.0% |
| OPEFB capacity | 2000 kg/batch |
| Labour wage | $0.92/h |

## 3. Results and Discussion

### 3.1. Kinetic Parameters of Fermentation

The compositional analysis showed that the un-pretreated OPEFB used in this study contained 39.0% cellulose, 29.8% hemicellulose, 22.8% lignin and 1.7% ash, according to Gozan et al. [24]. After being alkaline-pretreated, there were significant changes to the composition of the OPEFB, as can be seen in Table 3. The pretreated OPEFB contained 63.97% cellulose, 10.58% hemicellulose, 19.39% lignin and 1.51% ash. This result proved that the delignification of the OPEFB occurred during the pretreatment process. In general, alkaline pretreatment of biomass at elevated temperatures will dissolve hemicellulose and lignin effectively, as the degradation of hemicellulose occurs much

faster than that of cellulose [38]. This phenomenon can be seen from the decreasing amounts of both hemicellulose and lignin, but not of cellulose, in the pretreated OPEFB. The high cellulose content in the pretreated OPEFB would be very beneficial for the hydrolysis reaction to produce glucose during SSF.

**Table 3.** Composition of OPEFB after NaOH pretreatment.

| Parameters | Composition (%) |
| --- | --- |
| Ash | 1.51 |
| Lignin | 19.39 |
| Cellulose | 63.97 |
| Hemicellulose | 10.58 |

The effect of the fermentation time on the cell concentration at various fermentation temperatures can be seen in Figure 3b. Based on Figure 3b, cell concentration tends to increase with longer SSF times. This shows that the process of cell formation is still at an exponential stage because of the availability of sufficient substrate during the reaction and of reaction conditions that support the process of cell growth. Cell growth phenomena such as the lag phase, stationary phase and death phase have not been seen in the SSF process; this could be due to the large range of times for sampling processes. The lag phase in the cell formation can be predicted at $t \leq 24$ h. The exponential phase can be seen according to Figure 3b from the 24th h to the 72nd h, and after the 72nd h, cell growth looks constant or enters the stationary phase. On the other hand, the death phase was not seen in this study. This could be due to the abundant amount of substrate, which is indicated by the large residue of OPEFBs at the end of the SSF reaction ($t = 96$ h). The death phase was also not seen in a study conducted by Amenaghawon et al. in which it was stated that the death phase is not seen in the cell growth phase because the substrate is still available for microorganisms to carry out metabolic processes [39].

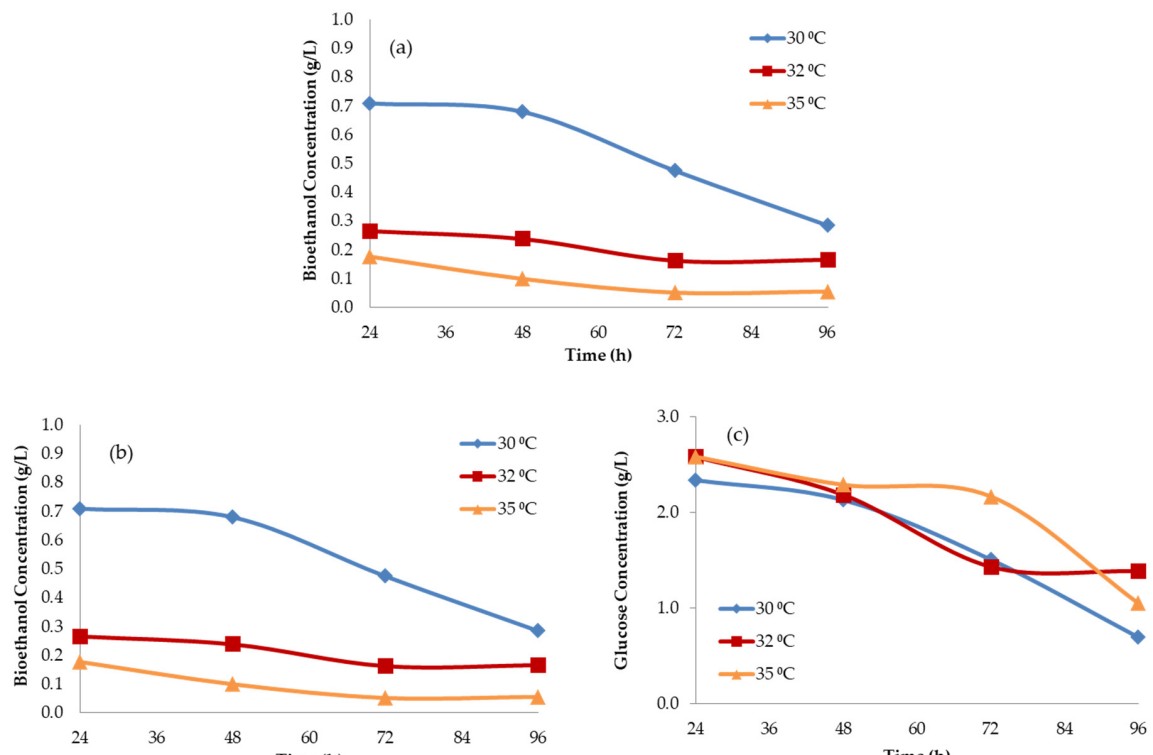

**Figure 3.** The concentration of bioethanol (**a**), yeast cell (**b**) and glucose (**c**) during SSF.

The effect of temperature on cell growth shows that the higher temperature of SSF produced higher microbial concentrations. This may have been due to the substrate that was used in this process.

The substrate used in this research was OPEFB, which must be broken down into glucose using the cellulase enzyme. The enzymatic hydrolysis process was generally carried out at a temperature of 50 °C ($T \geq 30$ °C). At 32 °C and 35 °C, the SSF process can cause the enzymatic hydrolysis of OPEFBs to be more effective and to produce more glucose than is produced at lower temperatures (30 °C). Glucose is used by yeast for cell growth.

The effect of the fermentation time on the glucose concentration at various fermentation temperatures can be seen in Figure 3c. Based on Figure 3c, glucose concentration tends to decrease with longer SSF times. This may occur because the glucose produced in enzymatic hydrolysis will be used directly by cells for growth processes and to produce bioethanol [39,40].

The decrease in glucose concentration showed the same tendency at each SSF reaction temperature. This indicates that in the SSF process, glucose that has been formed as a result of enzymatic hydrolysis will be used directly by yeast as a carbon source to form bioethanol. 30 °C was the temperature that showed the lowest glucose concentration at the end of the SSF reaction. This happened because, at this temperature, yeast cells make optimum use of glucose, which can be characterised by high ethanol concentrations.

The effect of the fermentation time on the bioethanol concentration at various fermentation temperatures can be seen in Figure 3a. Based on Figure 3a, the highest bioethanol concentration was 0.70 g/L. The longer reaction time of SSF produces lower bioethanol concentrations. The lower bioethanol concentration produced in this study was different from that produced in previous studies because, in general, the concentration of bioethanol will increase with longer fermentation times. The lower bioethanol concentration in this study could be due to the vaporisation of the ethanol.

The effect of temperature on bioethanol concentrations is quite significant. Figure 3a shows that a temperature of 30 °C with a 24-h reaction time produced bioethanol at around 0.70 g/L. At 32 °C and 35 °C with 24-h reaction time, the concentration of bioethanol dropped significantly to 0.26 g/L and 0.17 g/L, respectively. This might have occurred because higher reaction temperatures can make fermentation processes ineffective. High temperatures can affect cell transport activity and the occurrence of ribosomal and enzyme denaturation and can cause fluidity problems in cell membranes [41]. Yeast optimally ferments glucose into bioethanol at 30 °C. Based on data obtained from the laboratory, the bioethanol concentration was very small. Therefore, the influence of bioethanol as an inhibitor in the SSF kinetics equation can be ignored.

Based on Figure 4 above, the kinetic model optimisation of cell, glucose and bioethanol concentrations shows some results that are not optimal. This could be due to the composition of the raw materials used in this study. The raw material used in this study was OPEFB, which consists of cellulose, hemicellulose and lignin. The hemicellulose and lignin contents in the raw material could have inhibited the enzymatic hydrolysis process and the fermentation process. However, overall, the kinetics model that was used in this study is suitable for experimental data and can describe the SSF process.

From the results of the kinetic model optimisation with cell, glucose and bioethanol concentrations, various kinetic constants on the bioethanol reaction can be obtained. The kinetics constants for the bioethanol fermentation reaction can be seen in Table 4 below.

**Table 4.** Determined kinetic parameters for bioethanol production.

| Kinetic Parameters | Temperature (°C) | | |
|:---:|:---:|:---:|:---:|
| | 30 | 32 | 35 |
| $\mu$max (h$^{-1}$) | 0.009 | 0.013 | 0.018 |
| $ks$ (g/dm$^3$) | 0.004 | 0.010 | 0.025 |
| $kd$ (h$^{-1}$) | 0.009 | 0.009 | 0.213 |

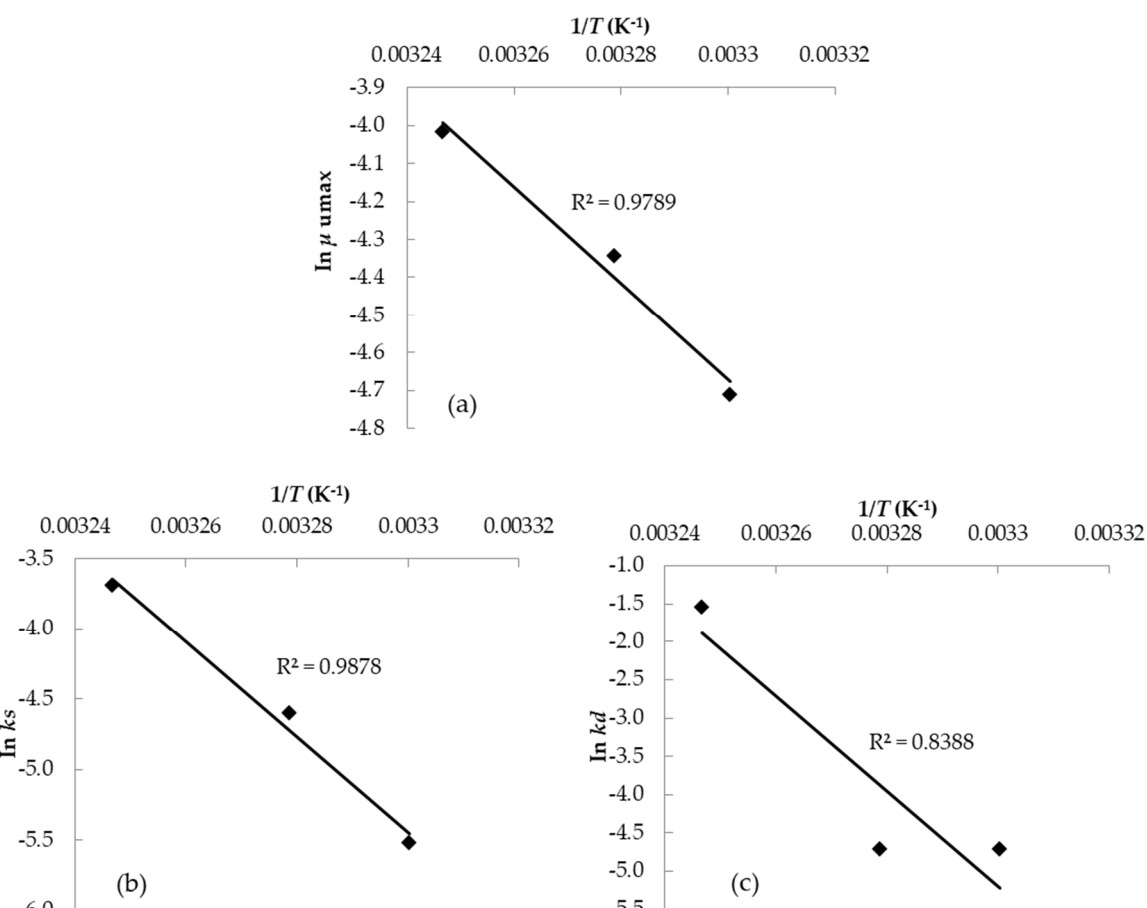

**Figure 4.** Arrhenius plot for SSF to determine $\mu$ max (**a**), $ks$ (**b**) and $kd$ (**c**).

Based on Table 4, it can be seen that higher temperatures will increase the value of various reaction constants, such as $\mu$ max (maximum specific growth rate), $ks$ (Monod constant), $kd$ (cell death rate constant) and m (maintenance cell constant). One parameter that influences cell growth is $\mu$ max. The effect of ethanol on cell growth can be seen through the $\mu$ max parameter. When the ethanol concentration rises, $\mu$ max tends to decrease. The same phenomenon is also seen in the previous research conducted by Amenaghawon et al. [39]. This indicates that bioethanol is an inhibitor of the cell growth process.

### 3.2. Process Simulation

The simulation result obtained using SuperPro Designer® v9.0 showed that fermentation time plays a significant role in determining the flowrate of products coming out of the fermenter when the kinetic parameters in Table 4 are applied. Figure 5 shows that the fermenter's output flowrate of bioethanol, glucose and yeast varied according to the fermentation time. The glucose flowrate decreased with increasing fermentation time as it was used as the substrate for bioethanol production. On the other hand, the bioethanol and yeast flowrates increased with increasing fermentation time, as they were both produced during fermentation. The glucose was completely consumed when the fermentation time was set to 192 h. At this time, the flowrate of produced bioethanol was at a maximum. Thus, the fermentation time for this hand sanitiser plant was set to 192 h to obtain the maximum bioethanol yield with the minimum batch time.

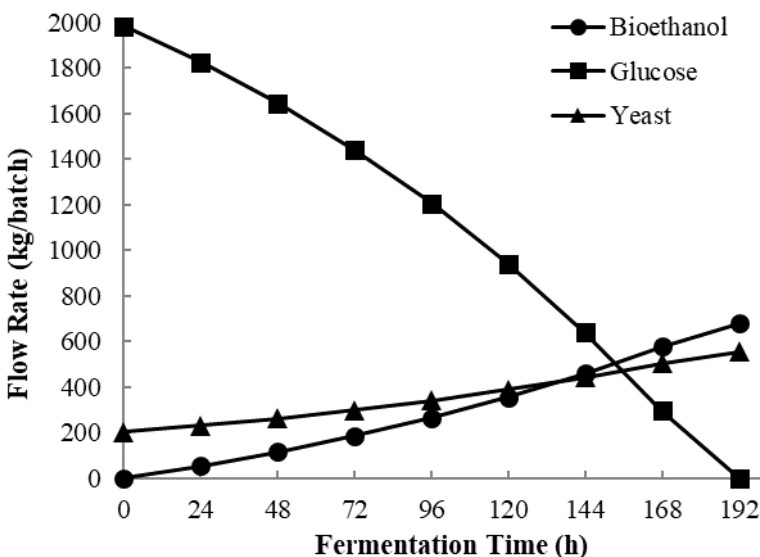

**Figure 5.** The relationship between fermentation time and flow rate of bioethanol, glucose and yeast leaving fermenter simulated by SuperPro Designer©. v9.0.

When the fermentation time was set to 192 h, the flowrate of produced bioethanol was 681 kg per batch, with a concentration of 3.4%. The plate and frame filtration could separate solid and liquid fractions of fermentation broth, resulting in an increased bioethanol concentration of 3.5% in the liquid output stream. The first distillation process in the beer column increased the bioethanol concentration to 79.3% with the remaining water. The second distillation process in the rectifying column finally increased the bioethanol concentration to 96.2%, as required by the WHO for use as a hand sanitiser raw material. At this point, the flowrate of bioethanol decreased to 651 kg due to some losses in the previous separation processes. Bioethanol was then mixed with the other raw materials, resulting in a final hand sanitiser product with a composition of 80.2% ethanol, 18.3% water, 1.4% glycerol and 0.1% hydrogen peroxide, as required by WHO standards. The overall results showed that this hand sanitiser plant design could produce 812.7 kg of hand sanitiser from 2000 kg of OPEFB for every batch.

The recipe scheduling information feature in SuperPro Designer® v9.0 for this hand sanitiser plant showed that the batch time and minimum cycle time (excluding equipment shared across batches and auxiliary equipment) were 219 and 194 h, respectively, with a total of 40 batches per year. This plant could, therefore, produce 32,506.16 kg of the total flow of hand sanitiser as the main product per year.

*3.3. Cost and Economic Parameters*

Investment performance measurement was conducted using several economic analyses, including the calculation of NPV, IRR, PBP and ROI. The calculations were performed using SuperPro Designer® v9.0 with several assumptions employed, based on the particulars of the Indonesian industry context, such as the number of holidays in a year, the labour price, the price of raw materials and the risk-free rate.

This section presents the estimation of the total capital investment cost (TCI), the annual operating cost (AOC) and the unit production cost of the hand-sanitiser product. The project is assumed to have a lifetime of 10 years and an annual operating time of 330 days (11 months). The depreciation of the capital investment was calculated using the straight-line method for a period of 10 years and a salvage value equal to 5% of the initial cost. The TCI of the plant was estimated according to the equipment purchase cost (EPC) following a well-established engineering methodology described in detail in the literature [42]. The equipment sizes were calculated using the simulation tool, based on the mass and energy balances. The EPC was estimated according to information from vendors, literature sources, and the SuperPro Designer® v9.0 equipment cost database. The total investment was estimated at $645,000; the breakdown of the TCI is presented in Table 5 below.

**Table 5.** Fixed capital estimate summary (2020 prices in $).

| Total Plant Direct Cost (TPDC) | |
| --- | --- |
| 1. Equipment purchase cost | 101,000 |
| 2. Installation | 33,000 |
| 3. Process piping | 35,000 |
| 4. Instrumentation | 41,000 |
| 5. Insulation | 3000 |
| 6. Electrical | 10,000 |
| 7. Buildings | 46,000 |
| 8. Yard improvement | 15,000 |
| 9. Auxiliary facilities | 41,000 |
| TPDC | 325,000 |
| **Total Plant Indirect Cost (TPIC)** | |
| 10. Engineering | 81,000 |
| 11. Construction | 114,000 |
| TPIC | 195,000 |
| **Total Plant Cost (TPC = TPDC + TPIC)** | |
| TPC | 520,000 |
| **Contractor's Fee and Contingency (CFC)** | |
| 12. Contractor's Fee | 26,000 |
| 13. Contingency | 52,000 |
| CFC | 78,000 |
| **Direct Fixed Capital Cost (DFC = TPC + CFC)** | |
| DFC | 598,000 |

The calculation of the annual operating cost (AOC) was conducted to determine the influence of operating components on the product cost. The calculated AOC for this plant was $305,000; its breakdown is summarised in Figure 6. As shown in Figure 6, the largest percentage of the AOC is facility-dependent costs, which consist of capital depreciation, maintenance, insurance and overheads [43]. This can be explained, as the high value of the TCI and the relatively short lifetime of the plant lead to higher annual depreciation. The second highest cost contributor is utilities, which consist of standard electrical power, steam (low pressure and high pressure), and a large amount of cooling and chilled water, with a total annual utility cost of $95,749. The highest contributor to the utility cost is electricity, at $58,807 annually, or more than 61% of the utility cost. This is a result of the long period of fermentation that leads to high electricity consumption for processes such as agitation and temperature control. Raw materials, despite the low price of OPEFB, contribute to 22% of the AOC. The breakdown of raw material and utility costs can be seen in Table 6.

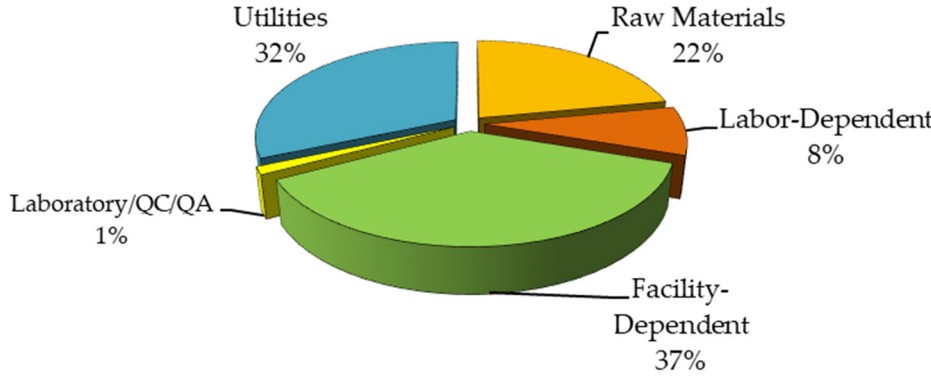

**Figure 6.** Annual operating cost breakdown.

**Table 6.** A detailed list of materials and utilities cost.

| Material/Utility | Unit Cost ($) | Annual Amount | Reference Unit | Annual Cost ($) |
|---|---|---|---|---|
| Air | 0.00 | 5,914,956 | kg | 0 |
| Cellulase | 2.50 | 4058 | kg | 10,146 |
| Citric Acid | 0.63 | 2866 | kg | 1791 |
| Glycerol 98% | 0.60 | 471 | kg | 283 |
| $H_2O_2$ 3% | 0.40 | 1356 | kg | 542 |
| OPEFB | 0.01 | 80,000 | kg | 488 |
| Sodium Citrate | 0.76 | 5431 | kg | 4127 |
| Sodium Hydroxide | 0.43 | 80,000 | kg | 34,000 |
| Water | 0.01 | 4600 | MT | 46 |
| Yeast | 1.95 | 8158 | kg | 15,907 |
| Standard Power | 0.10 | 588,067 | kW-h | 58,807 |
| Low Pressure Steam | 3.34 | 1983 | MT | 6624 |
| High Pressure Steam | 5.78 | 1119 | MT | 6468 |
| Cooling Water | 0.05 | 186,277 | MT | 9314 |
| Chilled Water | 0.19 | 78,573 | MT | 14,536 |

One of the reasons why this project could be economically beneficial is the low cost of labour in Indonesia. The actual minimum wage of labour in Indonesia varies depending on location. In this study, we used the highest value of minimum wage in Indonesia for labour and blue-collar jobs, which is about $0.9/labour-h. The annual cost for operators as labour in this plant was found to be $24,996 with a total annual amount of 27,170 h, and it contributes to about 8% of the AOC.

After calculating the TCI and AOC, the unit production cost of the hand sanitiser product was then calculated at $9.37 per kg of the main product. Since hand-sanitiser products are sold by volume, we determined the product selling price in USD/L. We varied the product selling price to further analyse the investment performance and to obtain the best-selling price. The calculation was performed using SuperPro Designer®, and the parameters we focused on were IRR, PBP, gross margin, ROI and NPV. We wanted to obtain the selling price most feasible for penetrating the market in Indonesia, most likely to be economically beneficial and most able to fulfil the standard economic parameters of the market. The detailed data pertaining to the effect of product price variation can be seen in Table 7 below.

**Table 7.** Effect of hand sanitiser price on some economic parameters in this study.

| Product Price (USD/L) | IRR (%) | PBP (Year) | Gross Margin (%) | ROI (%) | NPV (USD) |
|---|---|---|---|---|---|
| 2 | N/A | N/A | −295 | −26 | −1,824,000 |
| 4 | N/A | N/A | −97 | −14 | −1,278,000 |
| 6 | N/A | N/A | −32 | −3 | −732,000 |
| 8 | 0.4 | 10.9 | 1 | 9 | −224,000 |
| 10 | 9.8 | 6.1 | 21 | 16 | 108,000 |
| 12 | 17.0 | 4.3 | 34 | 24 | 440,000 |
| 14 | 23.2 | 3.3 | 44 | 31 | 772,000 |
| 16 | 28.7 | 2.6 | 51 | 38 | 1,104,000 |

*3.4. N/A: Not Available*

Based on the data mentioned above, it can be concluded that a product selling price of $10/L is the minimum selling price able to fulfil all the given parameters. The hand sanitiser product with a $10/L price produces a 9.8% IRR, will break even at 6.1 years, and has an NPV of $108,000 at the end of the plant's lifetime. With a gross margin of 21%, the product has a lower value according to the average gross margin obtained by speciality chemical industries, which have gross margin values at 31.16% [44]. Therefore, we need to choose the product selling price that can fulfil this parameter. The price point of $12/L produces a gross margin value of 34% with a higher IRR and NPV compared

to the $10/L price-point, with values of 17% and $440,000, respectively. It also has a relatively short payback period, at about 4.3 years. These economic parameters are from the market's point of view, with the average product selling price of hand sanitiser in Indonesia being $0.87 per 50 mL, or about $17.40/L, the product is economically competitive compared to the existing products in Indonesia.

## 4. Conclusions

This study explored the techno-economic evaluation of a hand sanitiser production plant using OPEFB-based bioethanol and the simultaneous saccharification and fermentation method. The kinetic parameters of bioethanol production using *Saccharomyces cerevisiae* in a laboratory-scale experiment were applied in SuperPro Designer® v9.0 to process simulation and economic assessment. The batch time of this hand sanitiser plant was 219 h, with a total of 40 batches per year. The total capital investment was calculated at $645,000 with a production capacity of 2000 kg per batch and a total of 32,506.16 kg of hand sanitiser as the main product per year. The total annual operating cost of this plant was found to be $305,000. The economic assessment performed by SuperPro Designer® v9.0 showed that the minimum selling price of the hand sanitiser was $10/L, which would result in a 9.8% IRR, a 21% gross margin, a 16% ROI, a $108,000 NPV and a PBP of 6.1 years. These results have shown that hand sanitiser production from OPEFB-based bioethanol is economically feasible and can be implemented at a tolerable price as an alternative application for second-generation bioethanol from lignocellulosic biomass in order to produce hygiene-related products.

**Author Contributions:** Conceptualization, M.G.; data curation, J.R.H.P.; formal analysis, A.F.P.H. and J.R.H.P.; investigation, J.R.H.P.; methodology, C.A.C.; project administration, M.Y.A.R.; resources, M.Y.A.R.; software, A.F.P.H.; supervision, P.S.; validation, P.S. and M.G.; visualization, A.F.P.H., C.A.C. and M.G.; writing—original draft, A.F.P.H.; writing—review and editing, P.S. and M.G. All authors have read and agreed to the published version of the manuscript.

**Funding:** We gratefully acknowledge the publication grant from Universitas Indonesia through Publikasi Terindeks International (PUTI) program Nr. NKB-1415/UN2.RST/HKP.05.00/2020, and partial support from MIT-Indonesia Research Alliance (MIRA) managed by Institut Teknologi Bandung (ITB) through WCU Program.

**Conflicts of Interest:** The authors declare no conflict of interest.

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
