# Peer review of "Techno-Economic Evaluation of Hand Sanitiser Production Using Oil Palm Empty Fruit Bunch-Based Bioethanol by Simultaneous Saccharification and Fermentation (SSF) Process"

_applsci, doi:10.3390/app10175987_

Round 1
Reviewer 1 Report
This is a timely and comprehensive presentation for the upconversion of an agricultural "waste" into a valuable product. Well presented.
The main question addressed by the research is the economic feasibility to produce ethanol (and then ultimately hand sanitizer) from a underutilized resource – open palm empty fruit bunch. The research is relevant – especially in the current climate with the global need for hand sanitizer – and is interesting because it explores the use of a second generation feedstock. The present manuscript adds to the current knowledge as often the economics of value-added products from lignocellulosic materials are not considered or are not included in reports. The authors have clearly shown that hand sanitizer produced in the manner stated could be profitable in Indonesia. The paper could benefit from adding the competitive nature to the abstract to emphasize the positive outcome of the research.
The paper is well written and presented. The conclusions are clear and align with the statements made in the manuscripts. The authors are clear when addressing the objective of the research.
Author Response
Dear Reviewer,
Thank you for your comments on our manuscript. We have revised our manuscript according to your comments as the following:
Comment:
The main question addressed by the research is the economic feasibility to produce ethanol (and then ultimately hand sanitizer) from a underutilized resource – open palm empty fruit bunch. The research is relevant – especially in the current climate with the global need for hand sanitizer – and is interesting because it explores the use of a second generation feedstock. The present manuscript adds to the current knowledge as often the economics of value-added products from lignocellulosic materials are not considered or are not included in reports. The authors have clearly shown that hand sanitizer produced in the manner stated could be profitable in Indonesia. The paper could benefit from adding the competitive nature to the abstract to emphasize the positive outcome of the research.
Revision:
Due to word limitation for the abstract, the authors have added a sentence “Until now, there have been no articles yet addressing the economic feasibility to produce value-added product like hand sanitiser, which is globally needed in current pandemic situation, from under-utilised resource like OPEFBs.” in the last paragraph of Introduction.
Reviewer 2 Report
This is a pretty well written manuscript that presents a timely and interesting research. After proper revision, it can be accepted.Below please see suggestions and comments that should help the authors improve this manuscript:
1) At the end of Introduction, the authors are recommended to add a phrase or two to specify what new and original this manuscript has. Not that the manuscript has no novelty and originality, but to highlight them and facilitate reading of the manuscript.
2) Figure 1. Optionally, if the authors agree, maybe the images need some scale bar?
3) All through the text: "hours" can be easily replaced with "h" which is a proper symbol for "hour(s)". Reducing text is a good idea as this text is a bit long.
4) Lines 317 and 324 (but also through the text, wherever it can be found): it is recommended to replace "gr/L" with "g/L". And "gr" with "g" wherever it was used in the manuscript.
5) Fig.3a-c, vertical axis: see comment 4) above: "gr/L" should be replaced with "g/L". For comparison, in Table 4, the authors use "g" rather than "gr"
6) Fig.6: optionally ,the authors may like to replace "dependent" with "related", if they agree that this term is more proper: Labor-Related & Facility-Related ?
7) Line 485, ref.6: what is the volume number of this article? 000 does not look like a proper number
8) Line 525, ref.21: what are the volume and page numbers of this publication?
Author Response
Dear Reviewer,
Thank you for your comments on our manuscript. We have revised our manuscript according to your comments as the following:
1) At the end of Introduction, the authors are recommended to add a phrase or two to specify what new and original this manuscript has. Not that the manuscript has no novelty and originality, but to highlight them and facilitate reading of the manuscript. Revision: The authors have added a sentence “Until now, there have been no articles yet addressing the economic feasibility to produce value-added product like hand sanitiser, which is globally needed in current pandemic situation, from under-utilised resource like OPEFBs.” in the last paragraph of Introduction.
2) Figure 1. Optionally, if the authors agree, maybe the images need some scale bar? Revision: We have added scale bar to the images in Figure 1A and 1B.
3) All through the text: "hours" can be easily replaced with "h" which is a proper symbol for "hour(s)". Reducing text is a good idea as this text is a bit long. Revision: We have replaced “hour(s)” with “h” all over the text in manuscript.
4) Lines 317 and 324 (but also through the text, wherever it can be found): it is recommended to replace "gr/L" with "g/L". And "gr" with "g" wherever it was used in the manuscript. Revision: We have replaced “gr/L” with “g/L” all over the text in manuscript.
5) Fig.3a-c, vertical axis: see comment 4) above: "gr/L" should be replaced with "g/L". For comparison, in Table 4, the authors use "g" rather than "gr" Revision: We have replaced “gr/L” with “g/L” in Fig. 31a-c.
6) Fig.6: optionally, the authors may like to replace "dependent" with "related", if they agree that this term is more proper: Labor-Related & Facility-Related ? Revision: The word “dependent” is commonly used in many plant design books for calculating operational cost thus we decided to keep this word.
7) Line 485, ref.6: what is the volume number of this article? 000 does not look like a proper number Revision: This is an accepted but unpublished article thus we have revised the reference number 6’s format by adding “(in press)”.
8) Line 525, ref.21: what are the volume and page numbers of this publication? Revision: This is an accepted but unpublished article thus we have revised the reference number 21’s format by adding “(in press)”.
|
Reviewer 3 Report
This reviewer is submitting an invited review of this manuscript.
The manuscript:
- deals with topics of high relevance,
- has scientific novelty and merit, and
- is written and presented with a high level of quality.
This reviewer would like to make the following comment and suggestion in regard to the sustainability of oil palm empty fruit bunch as raw material:
(manuscript lines 62-68: “Ethanol is commonly produced… household waste.”)
Comment:
- While the manuscript does not have a thematic focus on the sustainability of oil palm empty fruit bunch as raw material, the manuscript lines 62-68 nearly enter the highly-debated sustainability of the oil palm industry.
Suggestion:
- Given the challenges in the sustainability of the global oil palm industry (e.g. in connection to deforestation), despite important progress made in this industry, and keeping in mind the high share of the Indonesia production in the global production (as indicated in manuscript lines 98-100), it is suggested:
kindly consider inserting a few lines to indicate that it is advisable, maybe as part of further research, to consider conducting a comprehensive sustainability assessment whose scope would include the relevant on-farm practices when obtaining the oil palm empty fruit bunch as raw material.
This reviewer wishes the authors all success in their work.
Author Response
Dear reviewer,
Thank you for your comments on our manuscript. We have revised our manuscript according to your comments as the following:
Comments:
Given the challenges in the sustainability of the global oil palm industry (e.g. in connection to deforestation), despite important progress made in this industry, and keeping in mind the high share of the Indonesia production in the global production (as indicated in manuscript lines 98-100), it is suggested: kindly consider inserting a few lines to indicate that it is advisable, maybe as part of further research, to consider conducting a comprehensive sustainability assessment whose scope would include the relevant on-farm practices when obtaining the oil palm empty fruit bunch as raw material.
Revision:
We do concern about the issue in palm oil industry’s sustainability. Thus we decided to give a brief explanation on this in Introduction with the following: “As a demand and sustainability, palm oil serves many tasks such as excellent source of biomass, self-sufficient energy in processing, effective carbon sink, positive contribution to energy balance, sustainable practices and zero burning. The Indonesian Sustainable Palm Oil (ISPO) Certification Scheme is introduced by the Government of Indonesia as a mandatory requirement for all oil palm growers and mills to enhance the competitiveness of Palm Oil in the global market. For Malaysia, The Malaysian Sustainable Palm Oil (MSPO) Certification Scheme is the national scheme in Malaysia for oil palm plantations, independent and organized smallholdings, and palm oil processing facilities to be certified against the requirements of the MSPO Standards. The MSPO Certification Scheme provides the general principles for plantations and palm oil processing facilities to ensure that the palm oil products are produced in a responsible and sustainable manner [25].” and add a new reference for this.
Round 2
Reviewer 2 Report
The manuscript was properly revised and can be accepted for publishing